:::: PLOS | ONE

# Comparison of balance changes after inspiratory muscle or Otago exercise training

**Francesco Vincenzo Ferraro**[1]*, **James Peter Gavin**[2], **Thomas William Wainwright**[1,3], **Alison K. McConnell**[4]

**1** Department of Human Sciences and Public Health, Bournemouth University, Bournemouth, England, United Kingdom, **2** School of Health Sciences, University of Southampton, Southampton, England, United Kingdom, **3** Orthopaedic Research Institute, Bournemouth University Research Institute, Bournemouth University, Bournemouth, England, United Kingdom, **4** The Burrow, Christchurch, New Zealand

* fferraro@bournemouth.ac.uk

**Data Availability Statement:** All relevant data are fully reported within the manuscript. Additionally, a hard copy of the data is held within Bournemouth University facilities and can be made available under request.

## Abstract

The inspiratory muscles contribute to balance via diaphragmatic contraction and by increasing intra-abdominal pressure. We have shown inspiratory muscle training (IMT) improves dynamic balance significantly with healthy community-dwellers. However, it is not known how the magnitude of balance improvements following IMT compares to that of an established balance program. This study compared the effects of 8-week of IMT for community-dwellers, to 8-week of the Otago exercise program (OEP) for care-residents, on balance and physical performance outcomes. Nineteen healthy community-dwellers (74 ± 4 years) were assigned to self-administered IMT. Eighteen, healthy care-residents (82 ± 4 years) were assigned to instructor-led OEP. The IMT involved 30 breaths twice-daily at ~50% of maximal inspiratory pressure (MIP). The OEP group undertook resistance and mobility exercises for ~60 minutes, twice-weekly. Balance and physical performance were assessed using the mini Balance Evaluation System Test (mini-BEST) and time up and go (TUG). After 8-week, both groups improved balance ability significantly (mini-BEST: IMT by 24 ± 34%; OEP by 34 ± 28%), with no between-group difference. Dynamic balance sub-tasks improved significantly more for the IMT group (P < 0.01), than the OEP group and vice versa for static balance sub-tasks (P = 0.01). The IMT group also improved MIP (by 66 ± 97%), peak inspiratory power (by 31 ± 12%) and TUG (by -11 ± 27%); whereas the OEP did not. IMT and OEP improved balance ability similarly, with IMT eliciting greater improvement in dynamic balance, whilst OEP improved static balance more than IMT. Unlike IMT, the OEP did not provide additional benefits in inspiratory muscle function and TUG performance. Our findings suggest that IMT offers a novel method of improving dynamic balance in older adults, which may be more relevant to function than static balance and potentially a useful adjunct to the OEP in frailty prevention.

## Introduction

Physical activities have been shown to ameliorate age-related risk factors associated with falls [1]. In particular, multidirectional exercises (e.g. Tai Chi, Pilates or dance) have become

**Funding:** We also confirm that the project was part of (FF) PhD project and therefore received limited funding from Bournemouth University as part of (FF) PhD scholarship.

**Competing interests:** This work was sponsored by Bournemouth University. FF, JG and TW declare no conflicts of interests. AM acknowledges a previous (now expired) beneficial interest in POWERbreathe® inspiratory muscle trainers, in the form of a share of royalty income to the University of Birmingham and a potential share of royalty income to Brunel University. In the past, AM has also provided consultancy services to POWERbreathe® International Ltd., but no longer does so. AM is named on two patents relating to POWERbreathe® products, including the device used in the present study, as well as being the author of two books on inspiratory muscle training. The patents are as follows: European Patent Office Patent No. EP2303417B1 - Dynamic inspiratory muscle training device; United States Patent No. 8 459 255 - Dynamic inspiratory muscle training device. The authors also declare that the results of the present study do not constitute an endorsement by American College of Sport Medicine and that the results are presented clearly, honestly, and without fabrication, falsification, or inappropriate data manipulation.

popular to target balance deficiencies for older adults. Among these, the Otago exercise program (OEP) is an evidence-based intervention that is effective in reducing falls in older adults [2], as well as improving balance performance for both older healthy community-dwelling [3] and care home-dwelling adults [4].

The OEP involves group-based, lower-limb resistance (e.g. knee extension-flexion and hip abduction) and mobility exercises (e.g. tandem stance and walking) tailored to older adults who are at high risk of falling [5]. Kocic and colleagues [4] recently found that performing OEP three times a week, for 6 months, can improve dynamic balance (measured with the Berg Balance Scale) and physical performance (timed up and go and chair rising tests) in nursing home residents aged from 70 to 86 years. Although such multidirectional exercises may improve lower-limb strength and balance ability for older people, they require supervision, trained instructors and specific facilities, which can present barriers for many older adults.

In the last decade, alternative physical interventions have emerged, including those targeting the upper-body and trunk musculature [6] and in particular, inspiratory muscles (i.e. diaphragm and intercostal muscles), which have been shown to contribute to balance performance. During rapid limb movements, designed to perturb balance (i.e. shoulder abduction and adduction), the diaphragm is activated in a feedforward manner, assisting in the mechanical stabilisation of the spine [7]. In addition, inspiratory muscle contraction increases intra-abdominal pressure, which helps to stabilise the lumbar spine during static (e.g. standing on tiptoes) and dynamic (e.g. walking with head turns) movements that challenge balance [8].

Recently, with healthy community-dwelling older adults (73 ± 6 years) [9], we have shown that 8 weeks of unsupervised, home-based inspiratory muscle training (IMT) is both feasible and effective in improving balance ability. It improves dynamic and reactive balance, as well as gait speed and inspiratory muscle function for healthy older adults. However, the effectiveness of IMT *vs* an established falls prevention intervention, such as OEP is unknown.

This study compared balance and physical performance following either: i) daily, self-administered IMT with healthy community-dwelling older adults or, ii) instructor-led, group-based OEP with healthy residential care home-dwelling older adults. We hypothesised that, despite the different physical characteristics of the two groups, 8 weeks of home-based IMT would improve balance ability similarly to OEP.

## Methods

### Participant characteristics

Thirty-seven older adults (79 ± 7 years) participated in the study. Exclusion criteria comprised: aged under 70 years, acute respiratory tract infection, or chronic lung disease (e.g. asthma and obstructive pulmonary disease), having fallen in the previous 24 months, diabetes, any heart conditions preventing physical activity, taking beta-blocker medication, vertigo in the past 6 months, currently undertaking balance exercise training (including Tai Chi and Pilates) and previous experience of IMT.

Individuals with the following characteristics were also excluded: low balance confidence (activities balance confidence scale [ABC] lower than 67%) [10], moderate low back pain (Oswestry low back pain questionnaire [ODI] higher than 20%) [11], and cognitive impairments (Mini-mental examination test [MMSE] lower than 24) [12].

Participants were recruited through Bournemouth University public engagement events and residential care home managers in Dorset and Hampshire, UK (April to August 2018). Written informed consent was obtained by the principal investigator (FVF) before baseline measurements, and the research protocol was approved by Bournemouth University Research Ethics Committee (Reference ID: 19458).

## General design

A non-randomised repeated measures, pragmatic, parallel study design was used to investigate whether healthy community-dwelling older adults undertaking 8 weeks of unsupervised, home-based IMT would improve balance and physical performance outcomes, similarly to care home residents undertaking 8 weeks of instructor-led, group-based OEP. Nineteen (15 female, 4 male; age range 70–78) healthy community-dwelling older adults were allocated to IMT, and 18 (14 female, 4 male; age range 75–89) care healthy home-dwelling residents were assigned to OEP group classes.

This pragmatic non-randomised approach was selected for the following reasons:

1. The interventions were delivered according to the settings in which they were validated in to preserve internal validity. For example, IMT is effective as an unsupervised, individual home-based program, to improve balance and physical performance for older adults [9]. The IMT takes between 5 to 10 minutes per session. Conversely, the OEP is most effective as a group-based class [13] and involves supervision, as participants perform resistance and balance exercises, requiring monitoring of safety and technique. The OEP takes between 40 and 50 minutes per session.

2. To enhance external validity, the investigators aimed to reproduce a similar OEP group-based intervention as adopted in Geriatric Medicine (e.g. The Royal Bournemouth and Christchurch Hospital NHS Foundation Trust), whilst minimising financial and logistical costs for the participants. Residential homes having appropriate facilities, staff and equipment to perform the group-based OEP safely were recruited. The staff members of the care homes were recruited in an assistive role, to organise the facilities in which the OEP was delivered and to remind their guests about the appointments. Group allocation and participant pathways through the pragmatic study are illustrated in Fig 1.

Outcome measures were collected at the University premises for IMT participants, and at the care homes (six homes in total) of OEP participants. Testing and re-testing sessions were scheduled at a similar time of the day (between 8:00 and 11:00 am) to minimise potential effects of diurnal variation [14].

Participants were requested not to consume caffeine, alcohol or to perform strenuous exercise more than 2 hours before testing sessions. Assessments of forced vital capacity (FVC), forced expiratory volume ($FEV_1$), peak inspiratory flow rate (PIFR), peak inspiratory power, maximal inspiratory pressure (MIP), balance (mini-BEST), timed up and go (TUG), and 30 seconds sit to stand (30sSTS) were made at baseline (week 1) and post-intervention (week 8) by the principal investigator (FVF).

## Procedures

To enhance study reproducibility a step-by-step guide of the procedures is available at dx.doi.org/10.17504/protocols.io.8vhhw36

## Pulmonary function

A spirometer (SpiroUSB, Care Fusion, Wokingham, Berkshire, UK) was used to measure FVC and $FEV_1$ according to the American Thoracic Society guidelines [15]. Whereas PIFR was measured using the POWERbreathe® K5, with Breathe-Link 2.0 software (POWERbreathe® International Ltd, Southam, UK) using a technique validated by Langer and colleagues [16].

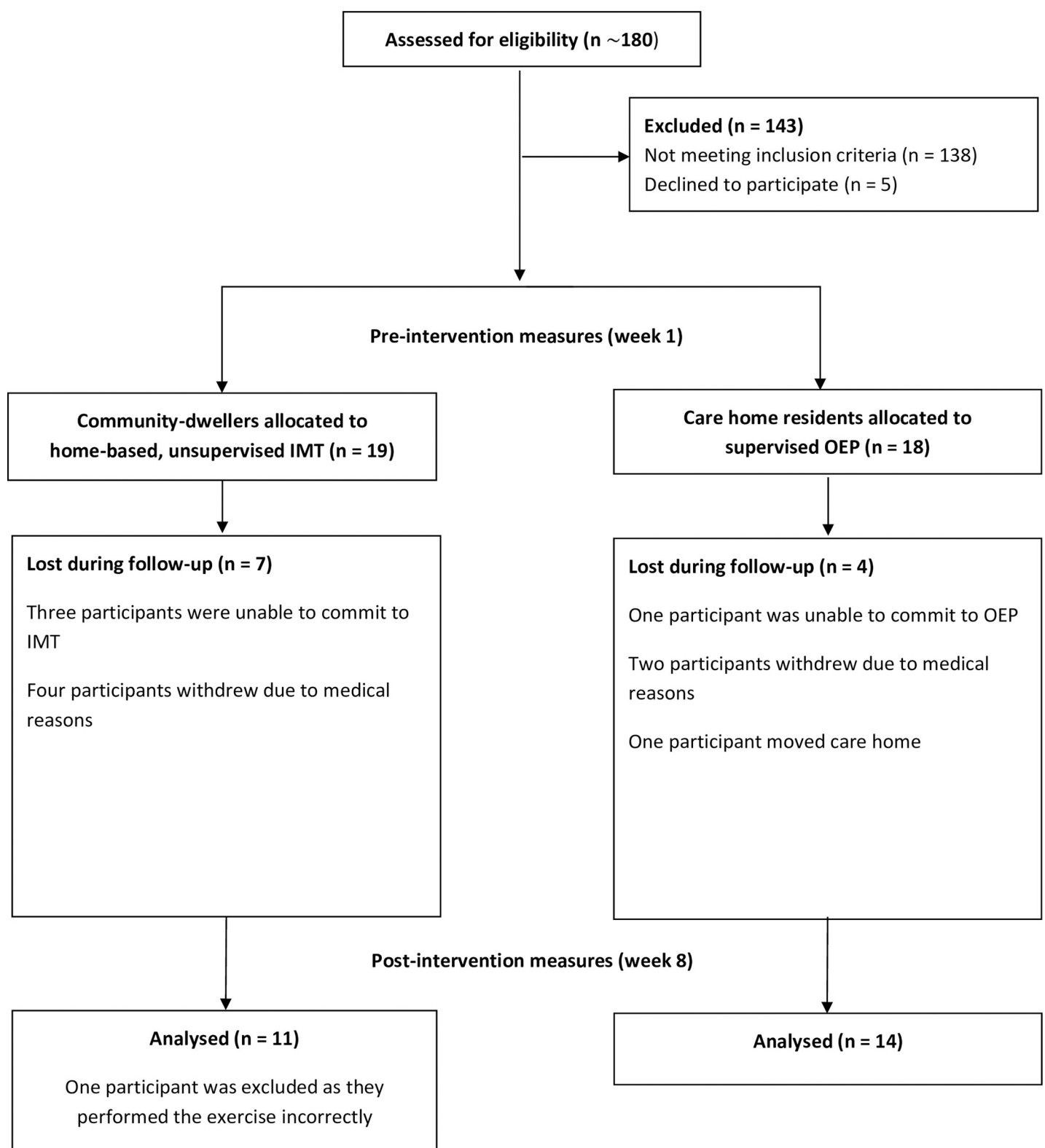

**Fig 1. CONSORT flow diagram displaying participant pathways through the 8 week non-randomised comparison study.** IMT = inspiratory muscle training, OEP = Otago exercise program.

Participants performed the forced breathing manoeuvres, at least five, and no more than eight times until variability was within 5% for three consecutive manoeuvres.

## Inspiratory muscle function

A hand-held mouth pressure meter (MicroRPM, Micro Medical Ltd, Rochester, Kent, UK) was used to determine MIP. The pressure meter was fitted with a side port opening of 1 mm internal diameter, to maintain glottis opening. All participants practised the Müller manoeuvre three times before testing, and MIP measurements were repeated, at least five, and no more than eight times, until variability was within 10% for three consecutive manoeuvres.

Peak inspiratory power was measured using the POWERbreathe® K5 with Breathe-Link 2.0 software [16]. Inspiratory muscle power analysis was undertaken at six different loads (40, 50, 55, 60, 70 and 80% of participants'MIP) and data were fitted with a polynomial curve, replicating methods reported elsewhere [17]. Participants were requested to inhale with maximal effort against the six loads, each of which was performed in random order (using randomizer. org software). Three trials were performed for each of the loading intensities, with 30 seconds rest intervals, making a total of 18 forced inspiratory manoeuvres.

Nose clips were worn for all pulmonary and respiratory muscle tests, and verbal encouragement was provided to promote maximal efforts. In each of the two respiratory muscle function measurements, the highest score was recorded.

## Balance

Static and dynamic balance were assessed using the mini-BEST, shortened version of the Balance Evaluation Systems Test, which included 14 different tasks, divided into anticipatory (e.g. rising on the toes for 3 seconds), reactive postural control (e.g. compensatory stepping forward), sensory orientation (e.g. standing on a foam surface with eyes closed for 30 seconds) and dynamic tasks (e.g. walking with horizontal head turns) [18].

## Physical performance

**The timed up and go tests.** The TUG was undertaken to assess mobility and gait speed. Two TUG variants were also used (i.e. the dual-task cognitive timed up and go [TUG$_C$] and motor timed up and go [TUG$_M$]) to measure the effect of dual-task on walking ability [19]. In all TUG assessments, participants were asked to sit on the edge of an armless chair (seat height 46 cm). On the command "3, 2, 1 and go" they were instructed to stand up, walk at their self-selected pace to a line on the floor 3 m away, turn around, walk back and sit down again [20].

During the TUG$_C$, participants were instructed to perform the same task, whilst counting aloud backwards in threes, from a randomly selected number between 80 and 100. During the TUG$_M$, participants completed the same tasks, as in the TUG, whilst holding a drinking glass (diameter 8 cm, height 9.5 cm) filled with water (1 cm away from the edge of the glass). Before testing, a familiarisation trial of the TUG task was performed. The three tasks were undertaken once in random order, with no physical assistance provided. Analysis to look at the added 'cost' of dual tasks to performance is reported as the difference [Δ] between the dual-task and single-task conditions.

**Sit to stand tests.** The 30 seconds sit to stand test (30sSTS) involved measuring the number of times participants were able to stand from a seated position, and then become seated again in 30 second period [21]. Briefly, participants were asked to sit on the edge of an armless chair (sitting height 46 cm, seat length 45 cm) with their arms folded across their chest. They were instructed to rise, and then become seated as fast as possible, and as many times as

possible in 30 seconds, with both feet maintaining contact with the floor at all times. Timing commenced on the command "3, 2, 1 and go".

The pre-activation 30 seconds sit-to-stand ($30sSTS_{PA}$) was used to determine the effect of acute, pre-inspiratory muscle activation [22] on the 30sSTS task. Briefly, adequate rest intervals between 30sSTS and $30sSTS_{PA}$ were provided (2 to 5 minutes), participants then performed 30 repetitions of forceful inhalations against a load equivalent to 50% of their MIP, followed by forceful inhalation at 80% of MIP until repetition failure. Verbal encouragement was provided during the 80% loading phase. When participants were unable to inhale forcefully for three consecutive attempts, they were instructed to stop the forced inspiration and to perform the $30sSTS_{PA}$. During both 30sSTS and $30sSTS_{PA}$ tests, participants were blindfolded to minimise a potential effect of vision on performance [23]. To further analyse the magnitude changes between sit to stand tasks the difference [Δ] between the standard condition and pre-activation condition are reported for both groups as 30sSTS vs $30sSTS_{PA}$.

**Trunk muscle endurance.**  Anterior and posterior trunk muscle endurance were measured only in the IMT group due to concerns about ensuring the participants' safety during this test in residential care homes.

Anterior trunk muscle endurance was assessed using an isometric 'sit-up' task, by adopting a bent knee (~75˚) sit-up. A strap secured participants' feet, their arms were folded across the chest, while their back was placed against a support (60˚ angle from the testbed), and knees and hips were flexed to 90˚. Participants were instructed to contract their abdominal muscles and maintain the position, whilst the support was manually pulled 10 cm back by the principal investigator [24].

Posterior trunk muscle endurance was assessed using the Biering-Sørensen test, where participants were asked to maintain a prone position, facing the floor, with their torso unsupported over the edge of the test bench. A strap secured their legs and hips, and hands were placed behind their head [25]. Participants were instructed to hold the static positions until the limit of tolerance, without verbal encouragement. Time was recorded using a digital stopwatch.

## Interventions

To improve the completeness of reporting, and the reproducibility of the interventions, both IMT and OEP are described according to the Template for Intervention Description and Replication (TIDieR) [26].

## Inspiratory muscle training (IMT)

Participants performed individual home-based IMT for 8 consecutive weeks, using a mechanical pressure threshold loading device (POWERbreathe Plus, POWERbreathe® International Ltd, Southam, UK). Participants followed an established training protocol known to improve inspiratory muscle function, which consisted of 30 breaths, twice-daily (once in the morning [between 7:00 and 12:00] and once in the evening [between 16:00 and 21:00]) at an adjustable resistance (equivalent to 50% of MIP). The IMT group's participants were instructed to increase the inspiratory resistance when they felt that 30 breaths were achievable with ease, or if they could reach 35 consecutive breaths [27].

## Otago exercise program (OEP)

The OEP consisted of a warm-up (10–15 minutes), resistance exercises with ankle weights (~20 minutes), balance activities (~20 minutes), and then a cool-down (5–10 minutes). The resistance exercises were tailored to the lower-limbs, including knee extension-flexion, hip

abduction and ankle plantarflexion-dorsiflexion movements. The OEP also involved exercises for the upper body, including head and trunk rotations and posterior neck hyperextension. The balance training exercises involved knee bends, backwards-, sideways-, stair-, heel and toe walking, tandem stance, the figure of eight, single-leg standing and sit to stand tasks. The classes were delivered by the principal investigator (FVF) in the residential home twice a week for 8 consecutive weeks, following guidelines of the Geriatric Medicine Department, The Royal Bournemouth and Christchurch Hospital NHS Foundation Trust (Fairmile Road, Christchurch, UK) (see S1 Appendix for the specific exercise regimen).

The group classes involved a minimum of two participants and a maximum of seven participants per class. The number of repetitions and the level of resistance were tailored to participants'physical fitness and progressively increased (by 0.5 kg if they were able to increase the number of repetitions by two). The resistance of each exercise was increased through ankle weights (ranging from 0.5–5 kg). Exercise sessions were conducted between 10:00 and 12:00. Participants were also instructed to walk for a minimum of 30 minutes at least twice a week (self-report), on the days in which the OEP was not performed. Attendance at the exercises classes was recorded by the principal investigator (FVF); participants were also instructed to cease exercising if they experienced any physical or mental discomfort.

## Data analysis

Sample size estimation was made using G*Power software [28] and data from our previous study involving 46 participants (73 ± 6 years) [9], using α error = 0.05 and 1-β = 0.95. To test our hypothesis that: healthy community-dwelling older adults undertaking 8 weeks of unsupervised home-based IMT, would improve balance outcomes to a similar magnitude to healthy care home-dwelling older adults undertaking 8 weeks of instructor-led group-based OEP, a total sample of 26 participants were required.

Comparisons were made using a repeated-measures ANOVA, with Bonferroni corrections to examine between-group effect magnitude (i.e. change after IMT *vs*. change after OEP), pre- vs post-intervention. Within group effects were examined using paired t-tests (i.e. pre- *vs*. post-intervention). Data are reported as mean, standard deviation (SD) and percentage change. The threshold for statistical significance was determined a priori as $P \leq 0.05$ and Cohen's d effect sizes were calculated to determine the effect magnitude (small $d \leq 0.2$; medium $0.2 < d \leq 0.8$; large $d > 0.8$). In addition, medium to large effect sizes were used to classify non-significant tendency.

## Results

Twenty-six participants out of 37 completed the study: adherence after 8 weeks was 63% for IMT and 78% for OEP. In addition, all participants undertaking OEP reported having walked at least twice per week (3.9 ± 1.8 days) for a minimum of 30 minutes. During the 8 week interventions, no participant reported adverse events (e.g. a fall accident or chest pain), and reasons for withdrawal were not related to the interventions. One IMT participant was excluded from data analysis, as they performed the exercise incorrectly (i.e. they forgot to increase the training load).

Both groups were similar at baseline for gender, body composition (BMI), cognition (MMSE) and lower back health (ODI < 20%). However, the IMT group was significantly younger (IMT: 74 ± 4 years; OEP: 82 ± 7 years; P = 0.03) and more functionally mobile (P ≤ 0.01), than the care home-dwelling OEP group (Tables 1, 3 and 4). Unexpectedly, after 8 weeks the ODI score increased for the IMT group (by 0.6 score points, d = 0.1; P = 0.008), but remained below the 20% minimal disability threshold (4.4%; Table 1). This increase appeared

**Table 1. Participant characteristics and pulmonary function at baseline (week 1) and post-intervention (week 8), following inspiratory muscle training (IMT, n = 11) and the Otago exercise program (OEP, n = 14).**

| Outcomes | IMT | | | OEP | | | P-values |
|---|---|---|---|---|---|---|---|
| | Baseline | Post-intervention | % change | Baseline | Post-intervention | % change | Between groups changes |
| Gender (M/F) | | 4/7 | | | 4/10 | | N/A |
| Age (years) [A] | | 74 ± 4 | | | 82 ± 7 | | N/A |
| BMI (kg m$^{-2}$) | | 27 ± 4.3 | | | 25 ± 5.3 | | N/A |
| ABC (%) [A] | 90.3 ± 9.7 | 92.0 ± 11.0 | 2 ± 18 | 67.1 ± 28.7 | 72.2 ± 27.9 | 8 ± 3 | NS |
| ODI (%) | 3.8 ± 6.4 | 4.4 ± 7.1 ** | 16 ± 11 | 7.5 ± 7.5 | 7.1 ± 8.8 | -5 ± 17 | NS |
| MMSE (total score 30) | 28.6 ± 1.0 | 28.6 ± 0.9 | 0 ± 10 | 27.9 ± 1.4 | 28.0 ± 1.4 | 0 ± 0 | NS |
| FVC (l) | 2.9 ± 0.9 | 3.0 ± 0.9 | 3 ± 0 | 2.2 ± 0.8 | 2.2 ± 0.8 | -2 ± 0 | NS |
| FEV$_1$ (l s$^{-1}$) [A] | 2.2 ± 0.7 | 2.2 ± 0.7 | 0 ± 0 | 1.6 ± 0.6 | 1.7 ± 0.7 | 6 ± 17 | NS |
| PIFR (l s$^{-1}$) | 4.9 ± 1.0 | 5.5 ± 1.1 | 12 ± 10 | 3.5 ± 1.3 | 3.6 ± 1.4 | 3 ± 8 | NS |
| MIP (cmH$_2$O) | 81.0 ± 24.1 | 134.4 ± 47.4 * | 66 ± 97 | 52.8 ± 31.1 | 61.0 ± 35.9 | 16 ± 15 | P = 0.001 |

Data are mean ± standard deviation, BMI = body mass index, ABC = activities specific balance confidence scale, ODI = Oswestry low back pain disability questionnaire, MMSE = mini-mental examination test, FVC = forced vital capacity, FEV$_1$ = forced expiratory volume in 1 second, PIFR = peak inspiratory flow rate, MIP = maximal inspiratory pressure. N/A = not applicable. NS = not significant.

* Significantly different from baseline (P ≤ 0.05),

** significantly different from baseline (P ≤ 0.01).

[A] = group were significantly different at baseline (P ≤ 0.01).

in Section 8 (i.e. *sex life*) for one participant only and for personal reasons not attributable to the effect of IMT.

## Pulmonary function and inspiratory muscle function

After 8 weeks, MIP increased significantly in the IMT group (66%; d = 1.4; P = 0.03), but not the OEP group (16%; P>0.05. Table 1). The magnitude of the increase was significantly larger for the IMT group (P = 0.001).

Fig 2 presents the interrelationships of inspiratory loads applied, and the inspiratory peak power and peak inspiratory flow rates generated. The magnitude of the increase in peak inspiratory power were significantly different (P < 0.01) between groups after 8 weeks at 50% and

**Table 2. Baseline (week 1) and post-intervention (week 8) values for peak inspiratory power at different percentages of load, following inspiratory muscle training (IMT, n = 11) and Otago exercises program (OEP, n = 14).**

| Peak power | IMT | | | OEP | | | P-values |
|---|---|---|---|---|---|---|---|
| | Baseline | Post-intervention | % change | Baseline | Post-intervention | % change | Between groups changes |
| 40% MIP (W) | 6.2 ± 3.6 | 6.9 ± 3.9 | 10 ± 8 | 2.9 ± 2.5 | 2.3 ± 1.5 | -23 ± 40 | NS |
| 50% MIP (W) | 5.9 ± 4.1 | 7.7 ± 3.6 ** | 31 ± 12 | 2.4 ± 1.9 | 2.2 ± 1.4 | -7 ± 26 | P = 0.01 |
| 55% MIP (W) | 7.2 ± 4.2 | 8.0 ± 3.4 | 11 ± 19 | 3.2 ± 3.5 | 2.4 ± 1.8 | -25 ± 48 | NS |
| 60% MIP (W) | 6.2 ± 5.2 | 7.8 ± 3.6 | 26 ± 30 | 2.6 ± 2.7 | 2.2 ± 1.8 | -14 ± 33 | NS |
| 70% MIP (W) | 6.6 ± 4.3 | 8.5 ± 5.0 | 30 ± 16 | 3.5 ± 4.3 | 2.2 ± 1.8 | -36 ± 58 | P = 0.04 |
| 80% MIP (W) | 6.3 ± 4.5 | 7.4 ± 3.6 | 17 ± 20 | 2.7 ± 3.2 | 1.7 ± 1.5 | -35 ± 53 | NS |
| MAX$_{PP}$ (W) | 8.9 ± 4.4 | 9.8 ± 4.0 | 9 ± 4.6 | 4.1 ± 4.1 | 3.3 ± 2.2 | 20 ± 46 | NS |

Data are reported as mean ± standard deviation, MAX$_{PP}$ = maximal peak power, MIP = maximal inspiratory pressure. W = Watts. NS = non-significant.

* Significantly different from baseline (P ≤ 0.05)

** significantly different from baseline (P ≤ 0.01). NS = not significant.

**Table 3. Baseline (week 1) and post-intervention (week 8) scores for balance and physical performance tests before and after training for inspiratory muscle training (IMT, n = 11) and Otago exercise program (OEP, n = 14) groups.**

| Outcomes | IMT | | | OEP | | | P-Values |
|---|---|---|---|---|---|---|---|
| | Baseline | Post-intervention | % change | Baseline | Post-intervention | % change | Between groups change |
| Mini-BEST [A] | 19.0 ± 4.1 | 24.2 ± 2.7 ** | 24 ± 34 | 14.6 ± 4.9 | 19.5 ± 3.5 ** | 34 ± 28 | NS |
| Anticipatory | 4.7 ± 1.0 | 5.3 ± 0.5 | 13 ± 50 | 3.6 ± 1.6 | 4.9 ± 0.9 ** | 36 ± 80 | NS |
| Reactive | 2.8 ± 1.9 | 5.0 ± 1.3 ** | 79 ± 31 | 1.9 ± 1.9 | 3.7 ± 1.1 ** | 95 ± 42 | NS |
| Sensory [A] | 5.2 ± 0.8 | 5.5 ± 0.7 | 6 ± 12 | 3.6 ± 1.7 | 4.4 ± 1.2 | 22 ± 29 | NS |
| Dynamic | 5.8 ± 1.8 | 8.5 ± 1.4 ** | 47 ± 22 | 5.5 ± 2.0 | 6.5 ± 1.7 | 18 ± 15 | P = 0.04 |
| 30sSTS (nSTS) | 13.2 ± 4.4 | 15.2 ± 5.1 | 15 ± 16 | 9.3 ± 4.6 | 10.8 ± 4.2 | 16 ± 9 | NS |
| 30sSTS$_{PA}$ (nSTS) | 13.4 ± 5.1 | 17.5 ± 6.3 † | 31± 23 | 10.0 ± 4.6 | 12.0 ± 5.6 | 20 ± 22 | NS |
| 30sSTS vs 30 sSTS$_{PA}$ (Δ nSTS) | 0.2 ± 1.6 | 0.8 ± 1.6 | | 0.7 ± 1.7 | 1.2 ± 2.3 | | NS |
| Sit-up (s) | 32.2 ± 27.2 | 56.4 ± 48.4 | 75 ± 78 | | N/A | | N/A |
| Biering-Sørensen test (s) (n = 9) | 69.8 ± 46.1 | 109.3 ± 66.7 | 56 ± 137 | | N/A | | N/A |

Data are mean ± standard deviation. The mini-BEST test has a maximum score (MS) of 28, and it is composed of four component Anticipatory MS 6; Reactive postural control MS 6; Sensory orientation MS 6; Dynamic gait MS 10. 30sSTS = 30 seconds sit to stand, 30sSTS$_{PA}$ = 30sSTS prior inspiratory muscles activation, nSTS = number of sit to stand completed.

[A] = group were significantly different at baseline ($P \leq 0.01$)

** Significantly different from baseline ($P \leq 0.01$)

† Significantly different from non-pre-inspiratory muscle activation condition ($P \leq 0.05$), NS = not significant. N/A = not applicable.

70% of [baseline] MIP (P = 0.01; P = 0.04, respectively) (Table 2). Within-group analysis showed that the IMT group significantly improved inspiratory power output at the 50% of [baseline] MIP load (1.79 W; 31%; d = 0.5; P = 0.02). Conversely, the OEP group tended to exhibit lower inspiratory power at all loads at post-intervention (Table 2 and Fig 2), although the group differences were not significant.

The maximal peak power (MAX$_{PP}$) for the IMT group at baseline was 8.9 ± 4.4 W, and occurred at 60% of MIP. Post-intervention, the MAX$_{PP}$ for the IMT group was 9.8 ± 4.0 W, and at 70% of [baseline] MIP. For OEP group MAX$_{PP}$ was 4.1 ± 4.1 W, and occurred at 70%

**Table 4. Baseline (week 1) and post-intervention (week 8) scores for TUG single and dual-task tests before and after training for inspiratory muscle trading (IMT, n = 11) and Otago exercise program (OEP, n = 14) groups.**

| Outcomes | IMT | | | OEP | | | P-values |
|---|---|---|---|---|---|---|---|
| | Baseline | Post-intervention | % change | Baseline | Post-intervention | % change | Between groups change |
| TUG (s) [A] | 8.9 ± 1.1 | 7.9 ± 1.4 * | -11 ± 27 | 16.8 ± 13.7 | 14.1 ± 11.0 | -16 ± 20 | NS |
| TUG$_C$ (s) [A] | 14.6 ± 7.2 † | 10.3 ± 3.2 † | -29 ± 55 | 23.4 ± 13.9 † | 19.9 ± 10.0 † | -15 ± 28 | NS |
| TUG$_M$ (s) [A] | 10.8 ± 1.6 † | 8.9 ± 1.5 ** † | -18 ± 6 | 20.0 ± 17.1 † | 17.8 ± 14.2 † | -11 ± 17 | NS |
| TUG vs TUG$_C$ (Δ s) | 5.7 ± 7.7 | 2.4 ± 2.5 | | 6.6 ± 6.7 | 5.8 ± 3.3 | | NS |
| TUG vs TUG$_M$ (Δ s) | 2.0 ± 1.2 | 1.0 ± 0.9 | | 3.2 ± 4.5 | 3.7 ± 3.7 | | NS |

Data are mean ± standard deviation TUG = timed up and go, TUG$_C$ = cognitive TUG, TUG$_M$ = motor TUG, TUG vs TUG$_C$ value (Δs) represent the differences between TUG and TUG$_C$ before and after intervention expressed in seconds, TUG vs TUG$_M$ value (Δ s) represent the differences between TUG and TUG$_M$ before and after intervention expressed in seconds.

* Significantly different from baseline ($P \leq 0.05$)

** Significantly different from baseline ($P \leq 0.01$)

† Significantly different from TUG ($P \leq 0.05$). NS = not significant. N/A = not applicable.

[A] = group were significantly different at baseline ($P \leq 0.01$).

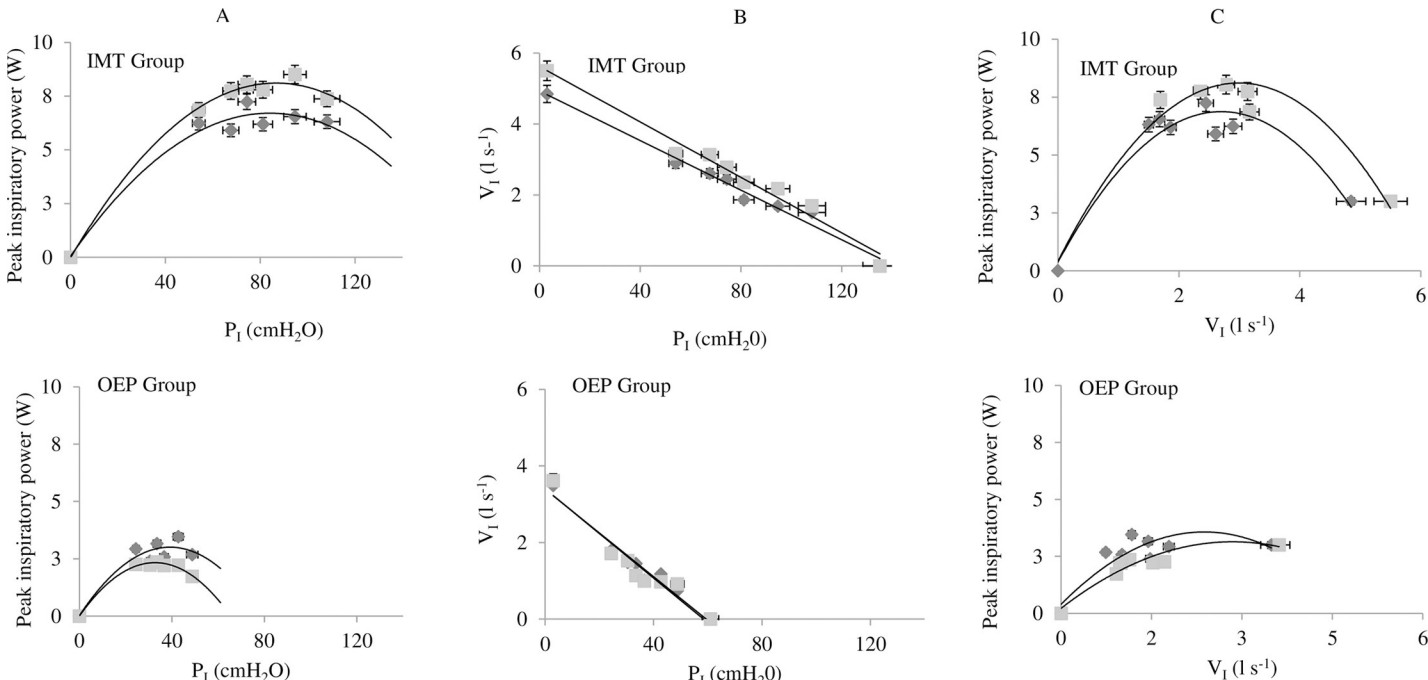

**Fig 2.** A. Peak power vs inspiratory mouth pressure $P_I$ (cmH$_2$O). B. Inspiratory mouth pressure $P_I$ (cmH$_2$O) vs inspiratory flow rate $V_I$ (l s$^{-1}$). C. Inspiratory peak power (Watts) vs inspiratory flow rate $V_I$ (l s$^{-1}$). Before (♦) and after (  ) 8 weeks of inspiratory muscle training (IMT) and Otago exercise program (OEP). Data are represented as mean ± percentage error.

MIP at baseline and 3.3 ± 2.2 post-intervention. The magnitude of the change between groups was not significantly different.

## Balance

The mini-BEST score was significantly different between groups at baseline (P ≤ 0.01. Table 3), and improved significantly following both IMT (by 24 ± 34%, +5 points, d = 1.2; P = 0.008) and OEP (34 ± 28%, +5 points, d = 1.5; P = 0.002). There was no significant difference in the magnitude of improvement between groups. Within-group analysis showed that the OEP group exhibited a significant improvement in specific sub-tasks; specifically in anticipatory tasks (d = 1.0; P = 0.01) and reactive tasks (d = 1.3; P = 0.02) (see Table 3), whereas the IMT group showed significant improvement in the reactive (d = 1.4; P = 0.004) and dynamic tasks (d = 1.7; P = 0.001). The magnitude of change in the dynamic balance tasks was significantly different between groups (IMT by 47 ± 22% and OEP by 18 ± 15%; P = 0.04).

## Physical performance and trunk muscle endurance

**The timed up and go test.** At baseline the IMT group performed significantly better than the OEP in TUG (IMT: 8.9 ± 1.1 s *vs.* OEP 16.8 ± 13.7 s; P ≤ 0.01), TUG$_C$ (IMT: 14.6 ± 7.2 s *vs.* OEP: 23.4 ± 13.9 s; P ≤ 0.01) and TUG$_M$ (IMT: 10.8 ± 1.6 s *vs.* OEP: 20.0 ± 17.1 s; P ≤ 0.01). However, after 8 weeks there was no significant difference in the magnitude of the change between groups (Table 4). Within group analysis showed that TUG and TUG$_M$ tests improved significantly in the IMT group (by 11%; d = 0.7; P = 0.02; by 18%; d = 1.2; P = 0.008, respectively). For both groups, TUG$_C$ and TUG$_M$ were significantly slower than TUG, both at baseline (P < 0.01) and post-intervention (P < 0.01). The magnitude of change of the dual-

task 'cost' on performance was not significantly different between or within groups and did not change as a result of either intervention.

**Sit to stand tests.**    After 8 weeks the only sit to stand test performance to show a significant change was the 30sSTS$_{PA}$, when compared with the baseline 30sSTS (by 15%; d = 0.4; P = 0.03. Table 3). There were no other within or between group differences.

**Trunk muscle endurance.**    Anterior and posterior trunk muscle endurance (i.e. sit-up and Biering-Sørensen tests, respectively) showed medium to large effect sizes, but no statistical significance. The sit-up test increased by 75% (d = 0.6, P = 0.12) and the Biering-Sørensen by 56% (d = 0.7, P = 0.20. Table 3). The OEP group did not perform trunk muscle tests (see Methods).

## Discussion

This is the first study to compare the effects of 8 weeks of IMT to those of OEP, upon balance and physical performance outcomes with healthy older adults. Our data confirm those of our previous study [9], showing that IMT improves inspiratory muscle function, balance and physical performance for healthy older adults; it is also consistent with findings for patients with COPD [29]. Our hypothesis that IMT and OEP would generate similar magnitudes of improvement in balance and physical performance was also confirmed for mini-BEST and TUG performances, despite the IMT group possessing superior function at baseline.

The IMT group improved in balance (assessed with mini-BEST) by 24 ± 34% and the OEP group by 34 ± 28%. Analysis of the four sub-components of balance showed that the IMT group specifically improved in dynamic balance task (e.g. walking with head turns) by 47 ± 22%, to a greater extent that the OEP group (18 ± 15%) after 8 weeks. Vice versa for static balance tasks (e.g. single-leg standing for 20 seconds) where IMT group increased by 13 ± 50% and the OEP group by 36 ± 80%.

The differences in training mode, frequency and dose (IMT: 5 minutes, twice daily for seven days per week *vs*. OEP: 40 minutes, once-daily for two days per week) influenced adherence, which was 63% for IMT and 78% for OEP. The 15% difference in adherence between groups concurs with the findings of Farrance and colleagues [30], who reported a mean adherence rate of 70% for group-based exercise programs with healthy community-dwellers.

Adherence to IMT was self-reported via training diaries, in which participants were instructed to record the number of breaths and the level of training load for each session (S2 Appendix). Adherence was verified with the MIP test, since MIP is expected to improve following IMT, due to adaptation to the overloading training stimulus [17]. Adherence in the OEP group was recorded by the principal investigator (FVF) prior to each training class.

### Pulmonary function and inspiratory muscle function

As expected, after 8 weeks, the IMT group improved MIP by 66% (d = 1.4), which is a larger improvement compared previous studies with healthy older adults (68 ± 3 years), after 8 weeks of IMT (34 ± 43%; d = 0.8) [31]. Also as anticipated, the increase in MIP tended to be accompanied by an increase in peak inspiratory power, which has been shown previously with young athletes [25]. The significant increase in power after 8 weeks of IMT, occurred at a load of 50% of baseline MIP, which corresponds to the training load. This result emphasizes the influence of training specificity upon adaptations to IMT [25]. The OEP group exhibited a slight deterioration in power output at all inspiratory loads, with the result that there were significant between-group differences in the power generated at loads corresponding to 50% and 70% of MIP. Accordingly, the IMT group improved inspiratory muscle function (force and power) after 8 weeks, but the OEP group did not.

## Balance

Balance ability was significantly different between the IMT group (healthy community-based) and the OEP group (healthy care home-based) at baseline. Despite this, after 8 weeks both groups exhibited similar improvements in the mini-BEST performance, i.e., 5 points, or around 30%.

The balance improvements shown by the OEP group were unsurprising, given that OEP is clinically effective in improving strength and balance [32]. Our finding is that IMT also improved balance, which is in agreement with our previous study, in which a similar cohort of healthy community-dwelling older adults (n = 23; 75 ± 6 years old), exhibited improvement in the mini-BEST performance (by 18%, d = 1.3) after an identical IMT intervention [9]. Our findings support our hypothesis that 8 weeks of unsupervised, individual home-based IMT for healthy community-dwellers and supervised group-based OEP for healthy older home-dwelling residents, would be similarly effective in improving balance ability. In addition, the mini-BEST sub-tasks demonstrated that the OEP group improved specifically in the 'static' anticipatory task (e.g. single-leg standing for 20 seconds), whereas the IMT group did not. Conversely, the IMT group improved in the dynamic task (e.g. walking at different speeds), whilst the OEP group did not.

These findings could help in isolating the possible physiological mechanism(s) by which each intervention improves balance. The OEP is known to improve the lower-limb strength of older women ($\geq$ 80 years) [33], and with participants who have a high risk of falls [5]. Therefore, gains in muscle strength and static balance from targeted exercises (e.g. standing on one leg) may underpin the improvements in the anticipatory task (i.e. standing on one leg).

**Potential mechanism 1 –upper-body and lower-body segmental linkage.** As mentioned in the introduction, the diaphragm has been shown to be activated in a feedforward manner (in conjunction with rectus abdominis), presumably to aid balance during rapid, destabilising movements of the upper limbs [7]. Greater inspiratory muscle strength, gained with IMT and measured objectively with MIP, may support the coordination of the upper-body and lower-body segmental linkage, enhancing the ability to increase intra-abdominal pressure by coactivation of the diaphragm and abdominal muscles [34].

**Potential mechanism 2 –intra-abdominal pressure.** Alternatively, the intra-abdominal pressure (IAP) has been established as a mechanism that improves spinal stiffness [8] with magnetic resonance imaging showing that IAP increases throughout diaphragm contraction, and during isometric lower-limb and upper-limb movement [35]. Therefore, it is rational to conceive that 8 weeks of IMT might produce improvements in dynamic balance tasks, as a consequence of improvements in the production of IAP.

The authors believe that to understand the potential relationship between inspiratory muscles activation and balance other measurements, such as EMG or intra-abdominal (gasti) balloon catheter, are necessary. Further research is required to investigate both, i) the efficacy of IMT, as a falls prevention intervention in frailer populations, and ii) the potential physiological adaptations conferred from an intervention tailored to improve inspiratory muscle strength, and balance ability.

## Physical performance and trunk muscle endurance

**The timed up and go test.** The IMT group showed significant improvements in TUG and TUG$_M$ performance by 11% (d = 0.8; P < 0.05) and 18% (d = 1.2; P < 0.01), respectively (Table 4). These findings agree with the results of our previous study with a slightly younger sample of older adults [9], where TUG performance was found to improve by 5.3% (d = 0.2). The improvement in TUG following our OEP intervention (16%; d = 0.2), though non-

significant, is double the percentage change observed following a similar intervention by Kocic and colleagues [4], who reported an 8% improvement (d = 1.0) after 3 months of OEP with care home residents (≥65 years).

It is plausible that the unpredicted absence of significant improvements in TUG performances for the OEP group is attributable to the wide age range of the participants (75 to 89 years) and therefore, to the heterogeneity of their mobility. In addition, at baseline the performance in the TUG, $TUG_C$ and $TUG_M$ tasks were different between groups (P ≤ 0.01), but not after 8 weeks of training. This supports our choice of a pragmatic, non-randomised research design and signifies that larger relative improvements occurred in TUG performance for the OEP group (TUG: IMT = 11% *vs* OEP = 16%).

Further analysis of single (TUG) and dual-task ($TUG_C$ and $TUG_M$) conditions showed that in both groups, the dual-tasks condition induced decrements in waking performance, pre- and post-intervention (Table 4), as participants needed more time to walk 6 m distance. Analysis of dual-tasks (TUG *vs*. $TUG_C$ and TUG *vs*. $TUG_M$) demonstrated that the dual-task 'cost' decreased following 8 weeks in both groups, with the exception of the $TUG_M$, which increased for the OEP group. The reason for this paradoxical deterioration post-intervention is unclear. The TUG findings support the abovementioned role of IMT in improving dynamic aspects of balance when compared to OEP, which specifically increased static balance ability.

**Sit to stand tests.** After 8 weeks, performance in 30sSTS tests showed a similar magnitude of improvement (Table 3) in both IMT and OEP groups (15% and 16%, respectively). Combining IMT with pre-activation of the inspiratory muscles enhanced performance in the $30sSTS_{PA,}$ when compared to that of the standard 30sSTS test, which is consistent with the results of our previous study with slightly younger healthy older adults (≥ 65 years old) [9]. After 8 weeks, participants of the IMT group were able to complete an additional two sit to stands in 30 s after pre-activation. The additive effects of IMT, plus inspiratory muscle pre-activation suggest the presence of an ergogenic "warm-up effect" upon the inspiratory muscles with older adults, similar to that reported with young adults [36].

**Trunk muscle tests.** Anterior (sit-up test) and posterior (Biering-Sørensen test) trunk muscle endurance were not affected by 8 weeks of IMT. However, the IMT group did show a moderate, though non-significant change post-intervention for both the sit-up (75%, d = 0.6) and Biering-Sørensen tests (56%, d = 0.7). The absence of significant changes is contrary to our previous study of a slightly younger sample of healthy older adults (sit-up: n = 23, 46%, d = 1.6; Biering-Sørensen: n = 22, 63%, d = 3.7) after an identical 8 weeks IMT intervention. The absence of a significant change in the present study is most likely due to insufficient statistical power.

## Limitations

The main limitation was the absence of random allocation, which meant that at baseline the IMT group was younger and had greater functional mobility (e.g. balance confidence and walking ability) than the OEP group of care home residents. These demographic and functional differences could have introduced an error in the absolute and relative changes observed, especially for the IMT group, where improvements might have been greater if a frailer cohort was recruited. However, to preserve validity (group-based exercises as adopted in current clinical settings) and feasibility (i.e. reduce travel and financial burdens) we were required to use care home facilities for the OEP group.

To our knowledge, this is the first study to compare an established physical training intervention, to a novel respiratory training intervention, which may influence mobility and dynamic balance. We were limited in delivering the IMT in care home settings, due to

heterogeneity in the health status of our population (e.g. a large proportion of residents have COPD and respiratory conditions).

We were also unable to employ blinded assessors or instructors, due to funding constraints to compare IMT against the OEP, in terms of improving participant outcomes in balance and physical performance. However, the principal investigator (FF) received competency training in physical training, respiratory muscle training, and technical aspects of all experimental equipment and underwent training at The Royal Bournemouth and Christchurch Hospital NHS Foundation Trust.

Concluding, due to difficulties in maintaining participant safety in care homes, we could not perform trunk muscle assessments (i.e. sit-up and Biering-Sørensen tests) with the OEP group. For future studies, we recommend measuring the effectiveness and feasibility of IMT on balance outcomes in frailer populations who are at a greater risk of falling.

## Conclusion

The findings of this pragmatic parallel study support our hypothesis that 8 weeks of unsupervised, individual, home-based IMT with healthy community-dwelling older adults, improves balance to a similar extent to supervised, group-based OEP with healthy older care home-dwelling residents. In particular, the results showed that IMT improved dynamic balance, whereas the OEP improved static balance for our older participants. Unexpectedly, the IMT group exhibited additional benefits to their walking speed (TUG, $TUG_M$) and inspiratory muscle function (strength and power), whereas the OEP group did not benefit in walking mobility (TUG). Further research is required to determine the potential mechanism(s) by which inspiratory muscles contribute to dynamic balance, as well as the role that IMT could play in improving balance proficiency as both a stand-alone intervention and as an adjunct to OEP and other falls prevention interventions. Concluding, the present findings suggest that IMT offers a novel method of improving dynamic balance in older adults, which may be more relevant to function than static balance and potentially a useful adjunct to the OEP in frailty prevention.

## Supporting information

**S1 Appendix. The Otago excercises program.**
(DOCX)

**S2 Appendix. Training diary used by the inspiratory muscle training group.**
(DOCX)

## Acknowledgments

The authors would like to thank all participants who volunteered their time to help with the research. We would also like to express great appreciation to the managers and the staff of Anchor Westmorland Court, Richmondwood Residential Home, Belvedere Court and Williams Court of Stonewater and Sunrise in Westbourne for their help and support.

## Author Contributions

**Conceptualization:** Francesco Vincenzo Ferraro, James Peter Gavin, Thomas William Wainwright, Alison K. McConnell.

**Data curation:** Francesco Vincenzo Ferraro.

**Formal analysis:** Francesco Vincenzo Ferraro.

**Investigation:** Francesco Vincenzo Ferraro, Alison K. McConnell.

**Methodology:** Alison K. McConnell.

**Project administration:** Francesco Vincenzo Ferraro.

**Supervision:** James Peter Gavin, Thomas William Wainwright, Alison K. McConnell.

**Writing – original draft:** Francesco Vincenzo Ferraro.

**Writing – review & editing:** Francesco Vincenzo Ferraro, James Peter Gavin, Thomas William Wainwright, Alison K. McConnell.

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
