## [Decision Letter · Decision Letter 0]

19 Sep 2019

PONE-D-19-18650

Comparison of balance changes after inspiratory muscle or Otago exercise training

PLOS ONE

Dear Dr Ferraro,

Thank you for submitting your manuscript to PLOS ONE. After careful consideration, we feel that it has merit but does not fully meet PLOS ONE’s publication criteria as it currently stands. Therefore, we invite you to submit a revised version of the manuscript that addresses the points raised during the review process.

We would appreciate receiving your revised manuscript by Nov 03 2019 11:59PM. To enhance the reproducibility of your results, we recommend that if applicable you deposit your laboratory protocols in protocols.io, where a protocol can be assigned its own identifier (DOI) such that it can be cited independently in the future. For instructions see: http://journals.plos.org/plosone/s/submission-guidelines#loc-laboratory-protocols

We look forward to receiving your revised manuscript.

Kind regards,

Joe Robert Nocera

Academic Editor

PLOS ONE

Journal Requirements:

3. Please include your tables as part of your main manuscript and remove the individual files. Please note that supplementary tables (should remain/ be uploaded) as separate "supporting information" files

'N/A'

Please provide an amended Funding Statement that declares *all* the funding or sources of support received during this specific study (whether external or internal to your organization) as detailed online in our guide for authors at http://journals.plos.org/plosone/s/submit-now.  Please state what role the funders took in the study.  If any authors received a salary from any of your funders, please state which authors and which funder. If the funders had no role, please state: "The funders had no role in study design, data collection and analysis, decision to publish, or preparation of the manuscript."

5.  We note that you have a patent relating to material pertinent to this article. Please provide an amended statement of Competing Interests to declare this patent (with details including name and number), along with any other relevant declarations relating to employment, consultancy, patents, products in development or modified products etc. Please confirm that this does not alter your adherence to all PLOS ONE policies on sharing data and materials, as detailed online in our guide for authors http://journals.plos.org/plosone/s/competing-interests by including the following statement: "This does not alter our adherence to  PLOS ONE policies on sharing data and materials.” If there are restrictions on sharing of data and/or materials, please state these. Please note that we cannot proceed with consideration of your article until this information has been declared.

Additional Editor Comments (if provided):

This manuscript examines the response of comparison of balance changes after inspiratory muscle training or Otago exercise training in community-dwellers. IMT and OEP improved balance ability similarly, with IMT eliciting greater improvement in dynamic balance, whilst OEP improved static balance more than IMT. Unlike IMT, the OEP did not provide additional benefits in inspiratory muscle function and TUG performance. Findings suggest that IMT may offer a new method of improving balance in older adults. The article is well written however there are serious concerns with blinding and objectivity considering the PI of the study conducted the assessments and administered the intervention. Why this is not a fatal flaw to the design will need to be described and highlighted in the limitations section. Statistical analyzes and their interpretation seem generally appropriate. These findings provide the first evidence that IMT improves balance and physical performance for healthy older adults compared to Otago exercise training.

Abstract- Please be more specific in introduction of the abstract for the term “community dwellers”. Are these older adults with balance impairment?

Introduction: The introduction describes the OEP as a successful approach to improve TUG however this study did replicate that finding. This should be discussed.

Methods: Participant characteristics- is it chronic obstructive pulmonary disease? How was it determined that patients had chronic lung disease? Self-report? If so, what about those that might have undiagnosed disease?

It is stated that “staff” were recruited to perform OEP. Were they required to have certifications and/or previous training? Were they trained to specifically perform OEP? It also stated that the “classes were delivered by the PI” but no details on training were given. This is of particular importance considering the OEP in this study does not match previous reports of improvements.

Seems alcohol should have been suggested to be avoided more than 2 hours before the exercise.

It is stated that the PI performed assessments. Because he/she delivered the intervention there are major concerns about objectivity.

More details on progression of the interventions is needed as well as added details on compliance to the interventions.

The results and discussion highlight “(12/19 and 14/18)” but it is unclear what these numbers are referring to.

The hypothesized mechanisms for noted improvements should be discussed/expanded.

Reviewers' comments:

Reviewer's Responses to Questions

**Comments to the Author**

1. Is the manuscript technically sound, and do the data support the conclusions?

Reviewer #1: Yes

2. Has the statistical analysis been performed appropriately and rigorously? 

Reviewer #1: Yes

3. Have the authors made all data underlying the findings in their manuscript fully available?

Reviewer #1: Yes

4. Is the manuscript presented in an intelligible fashion and written in standard English?

Reviewer #1: Yes

5. Review Comments to the Author

Reviewer #1: This manuscript examines the response of comparison of balance changes after inspiratory muscle training or Otago exercise training in community-dwellers. IMT and OEP improved balance ability similarly, with IMT eliciting greater improvement in dynamic balance, whilst OEP improved static balance more than IMT. Unlike IMT, the OEP did not provide additional benefits in inspiratory muscle function and TUG performance. Our findings suggest that IMT may offer a new method of improving balance in older adults. The experimental methodology was well designed using established physiological techniques and the article is well written. Statistical analyzes and their interpretation seem generally appropriate. These findings provide the first evidence that IMT improves balance and physical performance for healthy older adults compared to Otago exercise training. Excelent.

6. PLOS authors have the option to publish the peer review history of their article (what does this mean?). If published, this will include your full peer review and any attached files.

Reviewer #1: No

---

## [Author Response · Author response to Decision Letter 0]

14 Nov 2019

We thank all the reviewers for their insightful and constructive comments, which have allowed us to enhance the quality and clarity of the manuscript. 

We hope to have addressed the queries and concerns of the reviewers in the document Responses to Reviewers. 

Thank you 

best wishes

Francesco Ferraro.

---

## [Editor Report · Decision Letter 1]

18 Dec 2019

Comparison of balance changes after inspiratory muscle or Otago exercise training

PONE-D-19-18650R1

Dear Dr. Ferraro,

We are pleased to inform you that your manuscript has been judged scientifically suitable for publication and will be formally accepted for publication once it complies with all outstanding technical requirements.

With kind regards,

Joe Robert Nocera

Academic Editor

PLOS ONE

---

## [Editor Report · Acceptance letter]

26 Dec 2019

PONE-D-19-18650R1 

Comparison of balance changes after inspiratory muscle or Otago exercise training 

Dear Dr. Ferraro:

I am pleased to inform you that your manuscript has been deemed suitable for publication in PLOS ONE. Congratulations! Your manuscript is now with our production department. 

With kind regards,

on behalf of

Dr. Joe Robert Nocera 

Academic Editor

PLOS ONE